# How to Choreograph a Socialist Society?

Filip Petkovski

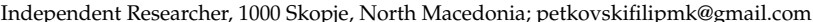

Independent Researcher, 1000 Skopje, North Macedonia; petkovskifilipmk@gmail.com

**Abstract:** During the existence of Yugoslavia (1945–1991), the leading political ideology of "brotherhood and unity" had to be manifested in all forms of cultural life. Promoting the physically capable body as part of a larger cultural movement, Yugoslavia witnessed the transformation of physical daily regimens into mass bodily spectacles performed at stadiums, called *sletovi*, demonstrating the power of mass-choreographed discipline. Similarly, Yugoslav choreographers were encouraged to develop a distinct performance aesthetic based on stylization as a rhetoric for modernization, using folk dance as a medium to showcase and promote the collective body of the people through choreographed folklore spectacles. Focusing on these two case studies that exemplify how mass choreography was used as a strategy to choreograph the Yugoslav society, this paper analyzes how political ideologies and their constructions through physicality supported the Yugoslav state project, thereby pointing to the present-day remnants of these aesthetics in the post-Yugoslav republics, evident in mass protests. By utilizing archival and choreographic analysis, I demonstrate how movement and dance impacted the public understanding of unity and helped the creation of a Yugoslav socialist society, drawing from Andrew Hewitt's thesis on "social choreography".

**Keywords:** Yugoslavia; socialism; social choreography; slet; kolo; folk dance; ideology; brotherhood; unity

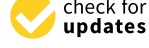



## 1. Introduction

The topic of choreographing society, as explored by several authors (Ness 1992; Olson 2004; Hewitt 2005; Rowe 2009; Giersdorf 2013; Petkovski 2023a), highlights the growing recognition of the profound implications on the performative nature of collective and mass bodily spectacle and their pivotal role in shaping our understanding of the complex relationship between body and politics. Such examples are also evident throughout the history of the Yugoslav state in several forms, which are the research subject of this study. The formation of the Yugoslav state was initiated around the idea of the unification of the South Slavs into one entity (Yugoslavia translates as the land of the South Slavs), especially with the conclusion of WWI and the dissolution of the Austro-Hungarian and Ottoman Empires. The first such attempts are evident in the formation of the Kingdom of Serbs, Croats, and Slovenes in 1918, later rebranded as the Kingdom of Yugoslavia in 1929. The transition from monarchy to socialism in 1945 led to a change in nomenclature, as the nation was renamed The Federal People's Republic of Yugoslavia in 1945 and, subsequently, The Socialist Federal Republic of Yugoslavia in 1963. Yugoslavia was a multi-ethnic and a multi-religious nation, comprised of six republics: Macedonia, Montenegro, Serbia, Bosnia and Herzegovina, Croatia, and Slovenia, along with two autonomous provinces, Kosovo and Vojvodina.

After the withdrawal from the Communist Information Bureau and the subsequent distancing from Soviet Union in 1948, President Josip Broz Tito initiated the non-aligned movement in 1961, bringing together leaders from newly decolonized countries and neutral states by providing a platform for member states to advocate for global peace, sovereignty, and development, while refusing to align with the superpower rivalries of the time (Stubbs 2023). What made Yugoslavia different from other socialist states was the distinctive path

of "soft socialism" or "liberal socialism" (Hofman 2011) and the accent on liberalization, as opposed to the Soviet-style collectivization existent in the countries of the Eastern bloc. The realm of popular culture within Yugoslavia occupied a unique position, straddling a historical lineage rooted in Eastern traditions, notably the Ottoman and Byzantine, while simultaneously displaying inclinations toward Western cultural trends. Politically, Yugoslavia adhered to socialist principles, yet it exhibited conspicuous consumerist tendencies akin to its capitalist counterparts (Čvoro 2014, p. 15).

In this work I trace how Yugoslav society was choreographed for the purposes of aligning it with the leading Yugoslav ideology of brotherhood and unity (*bratstvo i jedinstvo*), through the promotion of social and staged performances of dance heritage and the mass choreographed formations entitled *sletovi* (*slet* sing.). Some of the questions I aim to explore include the following: (1) In what manner does ideology perform itself through the utilization of the collective body of the people as its medium for embodiment? (2) How have performances of folk dances, as performed by state dance ensembles and amateur dance groups, embodied notions of Yugoslavism and togetherness? (3) To what extent have the Yugoslav performance aesthetics oriented towards the collective body of the people contributed to the establishment of a distinct socialist society? By utilizing choreographic analysis, I analyze a wide range of performances of folk dances and *sletovi*, as made available through personal archives, YouTube, and the archive of The Serbian Radio Television. Furthermore, I utilize archival analysis to examine archival video material of recorded performances of the Yugoslav *sletovi* in the 1970s and 1980s from The Serbian Radio Television and photographs from the Museum of Yugoslav History. I am guided by Foucault's genealogical approach (Foucault [1969] 1989) in order to counter the standard historiography that proves discontinuities when analyzing the ontology of the ideological matrix, which helps me excavate the genealogy of the collective body as a non-discursive historical formation.

My discussion is informed by prevailing theories on the concepts of choreography and ideology. While etymologically rooted in the transcription of movement into textual form, evidenced by historical dance notations (Franko 2015), choreography is additionally delineated as "a set of instructions for arranging the body in time and space, in patterns of stillness and movement, according to an established regime of techniques" (Banerji 2019, p. 31). The multifaceted applicability of choreography extends beyond traditional dance contexts, encompassing realms such as troop movements in the Iraq war, Cesar Millan's techniques in dog training, board meeting discussions, and the synchronization of traffic lights for enhanced commuter flow (Foster 2010, p. 60). Choreography extends beyond physical entities to encompass the structuration of movement, a concept delineated by scholars like Susan Leigh Foster (2010). In tandem with Foster's insights, Randy Martin (2011) identifies choreography as a locus of resistance, challenging established power hierarchies. The evolving discourse recognizes choreography's strategic utility in negotiating globality, hybridity, and local affiliations (O'Shea 2007), serving cross-cultural dialogues and cultural diplomacy globally (Croft 2015, p. 120). This role of choreography in configuring political structures and power dynamics (Lepecki 2013), extending beyond artistic expression, aligns with Susan Manning's (1993) critique of Eurocentric biases in historical choreographic discourse. Manning advocates for a more inclusive comprehension that embraces non-Western dance genres. Marta Savigliano (2009) contributes to this conversation by positing choreography as a strategic tool wielded by the West, transforming the mundane into dance through the capture of intrinsic movements.

Much akin to the dynamic exploration of choreography, the concept of ideology within critical discourse has undergone a profound and multifaceted examination. Karl Marx's foundational unveiling of ideology as a system of beliefs and values rooted in socio-economic structures (Marx [1867] 1992) initiated a theoretical discourse primarily associated with The Frankfurt School. Scholars such as Herber Marcuse and Theodor Adorno introduced a perspective that surpasses traditional Marxist and Freudian ideas, expanding on Marx's critique of political economy to encompass the broader concept of instrumental

reason. Following the Marxist tradition, Louis Althusser ([1974] 2014) advances the idea that ideology operates through ideological state apparatuses, including institutions like education and religion. Althusser underscores the pervasive role of ideology in sustaining social hierarchies. Building on these notions but adopting a post-structuralist perspective, Michel Foucault examines ideology as a discursive formation deeply intertwined with knowledge and power. Foucault exemplifies how ideology manifests in practices of governmentality, emphasizing its fluid and context-dependent nature (Foucault [1976] 2003). In the realm of political theory and philosophy, scholars like Laclau and Mouffe (1985) contribute to the conversation with their theory of hegemony and the role of discourse in shaping ideological formations.

These theoretical discussions have been crucial in analyzing the Yugoslav ideology of brotherhood and unity, a research subject of several authors (Djilas 1982; Ramet 1984; Silber and Little 1997; Glenny 2001; Malešević 2008; Jović 2009) who have emphasized the idea of transcending ethnic and religious divisions among the diverse population of the Yugoslav federation, according to an overarching objective to foster a sense of Yugoslav identity that would supersede narrower ethnic loyalties and create a unified, harmonious, and egalitarian Yugoslav society. The historical genesis of Yugoslav brotherhood and unity emerged in the aftermath of World War II, marked by Josip Broz Tito's leadership and the successful Partisan resistance against Axis occupation. The ideology was seen as a means to strengthen the federation, promote economic and social development, and prevent inter-ethnic conflict, all within the framework of a socialist system. However, it is important to add that this idea was initiated in the 19th century through the movements of Illyrianism and Yugoslavism, which were promoting ideas of the freedom of the South Slavs as a union of cultural groups that were perceived as outsiders and "other" (Ognjenović and Joželić 2016, p. 221).

The notion of "brotherhood" is also evident in the French revolutionary slogan of "Liberty, Equality, Fraternity" that conveys the idea of solidarity and unity among citizens striving for a democratic and egalitarian society. While both ideologies share the notion of fraternity, the French revolutionary context primarily emphasized the pursuit of liberty and equality alongside brotherhood. In contrast, Yugoslav brotherhood and unity specifically focused on fostering unity among diverse ethnicities within a socialist framework. Similarly, the Soviet concept of "Brotherhood of Peoples" highlighted the unity of various ethnicities within the socialist Soviet Union and aimed to portray a harmonious coexistence of diverse nationalities. However, the Yugoslav context was distinctive in its non-alignment during the Cold War and the unique model of socialism, which incorporated elements of self-management and cultural autonomy for different republics. Brotherhood and unity aimed to cultivate a collective identity and solidarity within a unified Yugoslav state, thereby mitigating historical animosities and fostering a sense of shared citizenship. The commitment to economic equality manifested in policies addressing socio-economic imbalances across regions and ethnic groups. The ideology's embodiment of non-alignment in foreign policy underscored Yugoslavia's commitment to socialist principles, advocating for peace and cooperation. This decentralized approach was intended to empower citizens and give them a stake in the system, thereby reinforcing the concept of brotherhood and unity. In practice, Yugoslav authorities promoted a shared Yugoslav identity through education, mass media, and cultural initiatives, including dances, as evident in this work, emphasizing the shared history of partisan resistance against Axis forces during World War II as a unifying narrative. The language policy also played a significant role in fostering unity, with the development of a common Serbo-Croatian language serving as a symbolic expression of Yugoslav integration.

In socialist Yugoslavia, the concept of "brotherhood" within the ideology of brotherhood and unity carried a complex gendered dynamic that exhibited masculinist aspects explored by several authors (Ramet 1998; Anđelković and Dimitrijević 1999; Dimitrov 2014; Bonfigioli 2018; Tumbas 2022). Even though the term "brotherhood" ostensibly conveyed a sense of camaraderie and unity among diverse ethnic and cultural groups within the

country, the gender aspect of this ideology was notably masculinist, reflecting prevailing societal norms centered around patriarchy and heteronormativity. Socialist and communist movements, both globally and within Yugoslavia, often used gendered language, so the choice of "brotherhood" was part of a broader linguistic tradition within socialist rhetoric that employed masculine terms to convey inclusivity without explicitly addressing gender equality. Furthermore, the leadership structure in Yugoslavia, as in many socialist states of the time, was predominantly male, so the use of gendered language may have been influenced by the male-dominated political environment, not meant to exclude women. The emphasis on masculinity, however, was accentuated in public performances as well, although it is important to add that they were not male-centered. For instance, the *sletovi* that I later analyze often featured more male than female participants, reinforcing traditional notions of masculinity associated with strength, discipline, and athleticism of the male body. Even though these performances served as public demonstrations of socialist values centered around unity in diversity, the embodiment of physical strength, particularly associated with men, became emblematic of the collective Yugoslav identity. Furthermore, dance heritage performances often emphasized traditional gender roles, evident in the divisions of male and female dances, distinctive gender roles where, in certain situations, men played the role of the leaders of the chain dances, aligned with the understanding that men are meant to showcase strength and speed, while women should embody grace.

Choreography, as explained in the following discussion, functions as an ideological vessel, carrying and manifesting ideological concepts through aesthetic means. A foundational assertion that instigates a dialectical discourse between choreography and ideology originates from Andrew Hewitt's work *Social Choreography: Ideology As Performance in Dance and Everyday Movement* (2005). In this treatise, Hewitt advances the concept of social choreography as the primary performative manifestation of contemporary social organization, establishing a critical nexus between the realms of aesthetics and the sociocultural domain. Consequently, choreography emerges as an instrumental medium for the rehearsal, configuration, delineation, or elucidation of social constructs and political ideologies within the aesthetic sphere. Inspired by Hewitt's work, several authors (Vujanović 2013; Cvejić and Vujanović 2015; Petkovski 2023a) have demonstrated how the Yugoslav state utilized choreography as a strategy to promote its distinctive form of socialism and the building of the Yugoslav socialist society. As shown through the case studies that follow, choreography has been employed not only for the purpose of achieving aesthetic ideals but also as a means of orchestrating and controlling large groups of people. Focusing on the dichotomy between choreography and ideology, I demonstrate Yugoslavia's successful attempts to choreograph its socialist society, drawing parallels of how these attempts are strongly inscribed within the cultural memory of the people, as evident in several cases of protest and political unrest.

## 2. Choreographing Folklore

Yugoslav folk dances have always played a central role, not only within Yugoslavia as a site of ideological context but also internationally, as a medium for achieving diplomacy and cultural branding. Framed as folklore, that is, the cultural production of the people transmitted through generations, where individual authors are unknown but rather authorship is collective, social participatory dances practiced in the villages of Yugoslavia gained momentum in the 1930s in the Kingdom of Serbs, Croats, and Slovenes, as they became the subject matter of the study of folklore and later, ethnochoreological research. Their categorization as folklore—a subject matter that would soon adopt political and ideological dimension—led to the organization of folklore festivals, which allowed for the possibility for social dances to be staged and publicly promoted, within the Kingdom and internationally (See, Petkovski 2023b). However, while village dance groups at times toured internationally, performing at various festivals, it was during the 1940s that a profound transformation took place, catalyzed by the establishment of cultural and artistic associations (*kulturno-umetnička društva*), which assumed the role of embodying socialist

principles of leisure as work through the execution of folklore, thereby setting in motion a significant shift in the manipulation and arrangement of dance patterns.

These organizations, often encompassing folk dance ensembles, choirs, drama, and sports divisions, were primarily conceptualized to propagate the concept of amateurism, wherein leisure was intended to be transformed into a productive endeavor, particularly within the context of nation-building and the cultivation of the new socialist citizen, whose values were to revolve around labor and productivity. Instead of seeking foreign cultural aesthetics to represent the national identity, Yugoslav authorities expressed a keen interest in promoting a sense of the common and the quotidian. By highlighting amateurism as a spontaneous collective expression, Yugoslav officials extolled the cultural contributions of the entire working populace. They regarded amateurism as a fundamental requirement for every individual seeking to become an integral part of the broader social community. Popular culture, encompassing music and dance, assumed a central role in cultural propaganda due to its association with populism. Consequently, cultural and artistic associations served as platforms that not only united the working class but also attracted the working intelligentsia, providing opportunities for public presentations of staged music and dance. The establishment of such associations underscored the notion that the production of culture necessitated a collective framework, manifested in the execution of social chain dances, which eventually evolved into symbolic representations of national culture.

These social dances, locally known as *kolo* or *oro*, represent a prevalent dance genre in the region, often translated into English as "circle dance" or "chain dance". Rooted in the Latin word "circulus", this *kolo* or *oro* dance formation serves as a unifying element not only among the South Slavic communities residing in the Balkan Peninsula (Mladenović 1973) but also in the cultures of various ethnic groups across Eastern Europe. Although the term ostensibly pertains to the cyclic spatial arrangement of dance, commonly featuring a circular or round configuration, *kolo* also conveys multiple connotations, encompassing a group of individuals engaged in dance, the dance event itself, and the specific dance style performed in harmony with a designated melody (Rakočević 2005, p. 1). The dance genre *kolo/oro* permeates all Yugoslav republics and neighboring Balkan nations, signifying an open or closed circle wherein participants, joined by their hands, execute synchronized steps, fostering a sense of equality and collective spirit. These dances offered an avenue for an unrestricted number of participants to come together, establish connections, forge bonds, and engage in collective interaction, all while immersed in the rhythms of the dance.

While the *kolo/oro* dances exhibit variations in names, rhythms, and tempos, a common characteristic is the facilitation of mass participation within circular formations, either open or closed, which may revolve in either a clockwise or counterclockwise direction, always advancing in the direction of motion but with frequent pauses or reversals. Dancers connect by joining each other's hands or shoulders, or in some instances, their sashes or belts, while in other instances intertwining their hands either behind or in front of their torsos. In social settings, particularly in rural areas, a discernible hierarchy dictates the positioning of dancers within the chain, with the first dancer holding a pivotal role as the leader who determines the course of movement and possesses the authority to influence tempo adjustments by signaling musicians to accelerate or decelerate. Similarly, the last dancer assumes a prominent position, engaging in communication with the leading dancer to modulate the flow of movement. In other instances where the dance is performed in closed circle, at a given time, certain individuals break from the circle and form a smaller circle within the bigger circle, sometimes moving in the opposite direction. There are instances when individuals leave the chain to perform a certain dance pattern in couples or solo but they always return to their place in the chain. Gender roles and hierarchies hold significant importance, with certain chain dances exclusively reserved for male or female participation, while others are mixed. In instances of mixed-gender participation, hierarchies often predominate, guiding the placement of individuals based on gender, family, or societal status (see Figure 1).

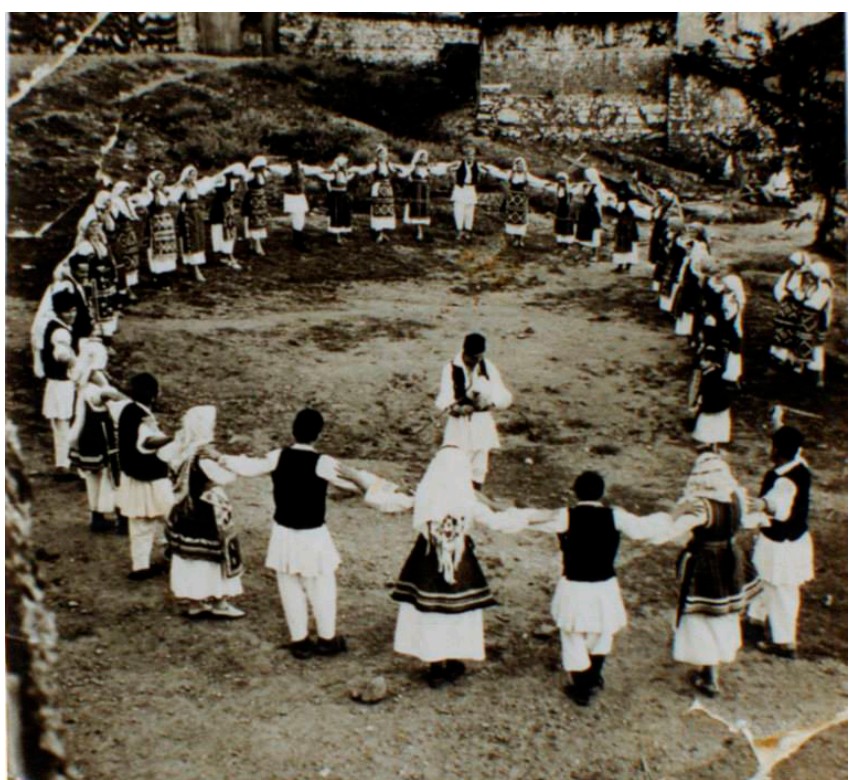

**Figure 1.** Dancers from the village of Rashtak in Macedonia, performing *oro* at the village square (source: personal archive).

Invariably, these dances are accompanied by live musical performances, ranging from solo instruments to full orchestras, typically situated either within the dance circle or on its periphery. Within the context of the dance event, the participants embody a representation of the collective community. Notably, no formal training is required to partake in these dances as proficiency is acquired solely through immersion. These types of dances are performed at all types of events, including religious holidays, weddings, social gatherings, and as shown in this paper, at protests as well. The participatory nature of the dance fosters inclusivity, allowing every community member to engage, thereby fostering a tangible sense of egalitarianism, irrespective of individual skill levels. All of these mentioned values held particular significance for Yugoslav ideologues, who adeptly incorporated them into their propaganda efforts, accentuating themes of unity, equality, and collective solidarity. A strategic shift involved the evolution from their communal origins toward orchestrated choreography intended for the stage, concurrently challenging the traditional concept of the solitary artist (Maners 1983, p. 12).

In the mid-1950s, Yugoslavia experienced a significant proliferation of the cultural and artistic associations—a development that necessitated the construction of additional performance venues, including *domovi kulture* (cultural houses), creating new opportunities for public performances, as well as a notable transformation of dance into a superstructural commodity serving the state's agenda. The objective of these performances, dedicated to local and national culture, extended beyond mere entertainment; they also aimed to educate the audience about the wealth of cultural diversity embedded in the repertoire and the importance of folklore as the product of the people's local cultures. In an effort to underline that culture and performance were not limited to the proscenium stage, various art forms, including drama, music, and dance, were brought to workplaces such as factories and other labor-intensive settings, where the working population of Yugoslavia spent a significant portion of their time. Consequently, many factories, universities, and other labor institutions established their *kulturno-umetničko društvo* that featured choirs, drama, and dance sections, thereby fostering a vibrant cultural scene within these workspaces.

The popularization of cultural and artistic associations gave birth to the idea of forming the first national and professional dance ensembles, whose mandate was to gather, adapt, and preserve the folk dances and songs of their respective regions. In Yugoslavia, the first such ensemble was the Serbian "Kolo", founded in 1948, followed by the establishment of the Macedonian and Croatian national dance ensembles "Tanec" in Skopje and "Lado" in Zagreb in 1949. However, these institutions that were to act as the cultural ambassadors of Yugoslavia had to adopt a certain process of modernization when it came to the choreographed material, aimed at elevating these dances from their rural origins to the realm of "high culture". This creative liberty and the overarching objective of modernizing folk culture encompassed a process known as *stilizacija*[1] (stylization), which entailed modifications to the music and dance elements, effectively tailoring them as staged spectacles. The intent behind these changes was to contemporize the performances, shedding their rural character and aligning them more closely with the aesthetic norms prevalent in Western culture.

However, the process of choreographing dance heritage, as well as what choreography is in the folklore realm, differed from the Western understanding outlined in the introduction of this piece, as it does not pertain to the creation of entirely new dance works but instead describes the process of organizing and structuring pre-existing forms of social dance and the integration with new step patterns. These changes encompassed modifications aimed at adhering to established aesthetic standards, which entailed shortening the dance's duration, altering its pattern, and adapting it for stage presentation, ensuring dancers remained constantly visible to the audience. The emphasis shifted from improvisation to performing uniform collective movements, with each dancer synchronizing their movements with the rest. Choreographers typically selected dances with more complex and visually captivating structures and dance steps, discarding parts deemed too simple and lacking audience appeal. Many choreographers found the repetitive nature of step patterns performed in the chain dance as monotonous and sought to revamp the choreography by introducing innovative movements, complementing the dance's fundamental motifs. In their endeavors to stylize and modernize dance performances, choreographers heightened the tempo, exaggerated movements, introduced acting elements, included polyphony and choral song arrangements, introduced foreign musical instruments of Western origin, and integrated narratives uncommon in traditional social dance presentations. These modernization principles were soon adopted by the amateur dance ensembles and established a specific aesthetic of dance heritage performance that was adopted on a national level, with the exception of some Croatian and Slovenian dance ensembles who strived to showcase authenticity in their performances by mimicking social dance performances, as observed in their villages where they originated.

Cognizant of the potential for folk dance performances to catalyze separatism and nationalism, Yugoslav authorities advocated for the creation of a pan-Yugoslav repertoire, encompassing dances and music from all the diverse ethnic groups and republics within the country. This strategic decision, rooted in the principle of brotherhood and unity, served as a proactive measure aimed at averting ethnic, religious, and political discord. Within their music and dance ensembles, Yugoslav dancers and musicians were required to acquaint themselves with the music and dance cultures from other republics, while local village performing groups were exempted from the pan-Yugoslav program as they were regarded as the authentic custodians of folklore traditions. Additionally, this diverse repertoire offered dance groups the opportunity to showcase an array of ethnically distinct material in order to forge a concept of national Yugoslav culture and cultivate a sense of Yugoslav identity, as well as a sense of unity and multiculturalism. Hence, choreography was perceived as an effective instrument for cultural propaganda capable of shaping a novel Yugoslav identity (Čvoro 2014, p. 39), while dance performances would assume the role of Yugoslavia's prominent cultural exports, essential for the country's international branding efforts.

The instances illustrating the development of the pan-Yugoslav program exemplify the tangible manifestation of the brotherhood and unity ideology in the realm of choreographic

production. However, this manifestation primarily emerges in the realm of program selection rather than the choreographing process per se. The choices in repertoire would not have materialized without common elements found in the local repertoires of the communities spanning all Yugoslav republics, as exemplified by the prevalence of chain dances. These strategic repertoire selections served the purpose of endowing dance participants with a sense of Yugoslav identity. It is noteworthy, however, that dance alone did not autonomously generate a sense of identity; rather, such identity perceptions were shaped by state influence, which elevated dance to the status of folklore—a category of utmost national significance and a marker of modernization. These initiatives bear resemblance to Andrew Hewitt's analysis of the nineteenth-century transformation in cultural interests, where the concept of choreographed labor assumed a pivotal role, serving as a significant element in both the process of social modernization and the aestheticization of social and political ideologies (Hewitt 2005, p. 38). The shift toward the popularization of folklore is particularly evident in the concerted emphasis on folklore production and promotion as a form of mass entertainment. This endeavor formed an integral component of the broader Yugoslav agenda aimed at modernizing local culture and positioning it as a national asset.

In these instances, choreography serves not only as a strategy embodying ideology but also as a mechanism for the political redemption of artistic activity in a Rancièrian sense. Chain dances such as *kolo/oro*, with their circular formations that preclude individualism, racism, and class distinctions, underscore the intrinsic connection between ideology and performativity. This notion aligns with Derrida's perspective, which posits the performative as a form of communication extending beyond the conveyance of pre-established semantic content, transcending the dominant orientation toward truth (Derrida 1982, p. 13). In the context of *kolo/oro* dancing, ideology takes on an active role, not merely signifying but actively generating a specific reality. This happens as the act of performing dances passed down through generations places dancers in a position of passive consumption rather than creative agency. During the moments when dancers find themselves in a liminal state, straddling the boundaries of distinct individuality and collective embodiment, their identities, intertwined with the characters they portray, become discernible within transitional realms characterized by "characterization", "representation", "imitation", "transportation", and "transformation" (Schechner 1985, p. 4). Of particular significance in the context of the ideological apparatus of unity and togetherness is the enduring nature of the fundamental meaning embedded in these chain dances. Such examples point out how performance conveys national significance and pride by elevating the social and communal aspects of dance to the status of vessels for enacting theories and ideologies employed by the Yugoslav state, as further exemplified in the following section. This interplay between choreography and ideology in the realm of dance underscores the intricate relationship between political narratives, cultural identity, and embodied expression, offering profound insights into the performative nature of society itself.

## 3. Choreographing the Collective Body

A prominent aspect of the commemoration of one of Yugoslavia's most prominent holidays, Youth Day, revolved around the grand-scale performances of *kolo/oro* (see Figure 2). During this celebration, numerous groups of dancers, each attired in traditional costumes representing various Yugoslav republics, partook in mass chain dances as an integral component of the festivities. These mass dance displays constituted just one segment of a larger spectatorial event known as the *slet*, which translates to the "flocking of birds". This annual occurrence unfolded at the venue of the Stadium of the Yugoslav People's Army (referred to as JNA) in Belgrade, coinciding with the conclusion of a Relay baton race (*štafeta mladosti*). The Relay baton race was a ceremonial and mass-participatory running event, wherein tens of thousands of individuals, comprising the Yugoslav working population, relayed a baton containing a birthday message for President Tito across all the republics. The culminating act transpired on 25 May, the day President Tito proclaimed as Youth

Day in 1956. Consequently, the JNA stadium, from 1956 onwards, became the designated location for the annual *slet* festivities.

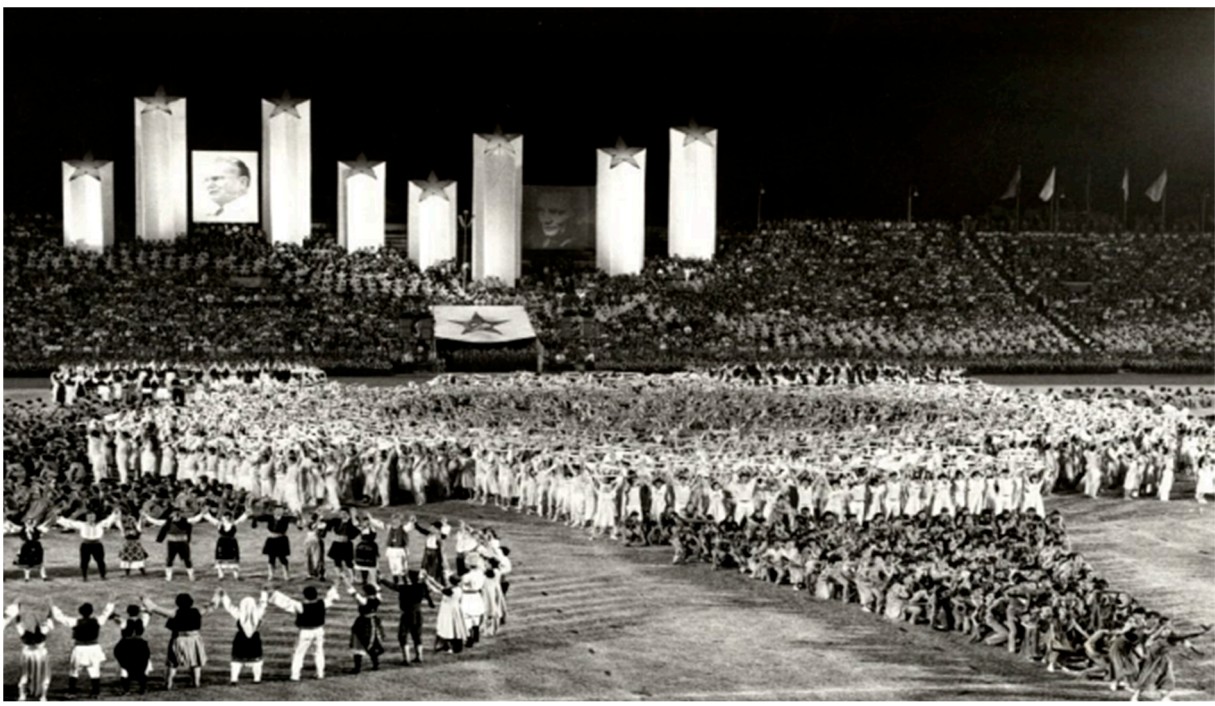

**Figure 2.** *Slet* performance in 1979 that features a *kolo* performance in the lower left corner (source: Museum of Yugoslavia, Photo 1979_697_156).

These celebrations persisted even after President Tito's death, extending until 1988, two years prior to the disintegration of Yugoslavia. It is noteworthy that the inception of Youth Day coincided with the promotion of Yugoslav self-management principles, and the mobilization of the masses through labor allegorically symbolized the operational ethos of Yugoslav self-management (Jakovljević 2016, p. 81). Other than dance, the *sletovi* included other elements such as gymnastic displays and military parades, utilized not only as artistic expressions but also ideological tools, as they were used to reinforce the values of socialism and Yugoslav unity. They captured the intricate interplay between culture, politics, and ideology, encapsulating the multifaceted character of the Yugoslav project and its aspirations for social and political cohesion amidst a complex and diverse national landscape.

The performance and organization of *sletovi* were tied to the *Sokol* (falcon) movement that originated in Czechia in 1862, initiated by philosopher Miroslav Tyrš as an opportunity to celebrate and promote notions of physicality as a cultural movement amongst the Slavs but also as a tactic aimed to oppose the Austro-Hungarian and Ottoman occupation of Central and Southeastern Europe. Within the Yugoslav area, the Sokol movement became prominent during the reign of King Aleksandar of the Kingdom of Serbs, Croats, and Slovenians (Zec 2015), and his son, King Petar II Karađorđević, who invested in the construction of gymnasium halls that would accommodate such events (Vujanović 2013, p. 21). Some authors (Nolte 2002; Troch 2019) have recognized the ideological matrix of orchestrating and choreographing the collective body through massed choreographic spectacles as a strategy aimed to produce a community centered around the physically healthy and strong and abled body. However, the ideology of brotherhood and unity, specifically, was rooted in the idea that such formations allowed for the overcoming of class differences and individualism, which posed a threat to the system.

The prevailing concepts of solidarity, counter-elitism, antifascism, and antinationalism found their fitting expression in the celebration of Yugoslavism and socialism, effectively

exemplified in the Youth Day performances. This event additionally served as a vehicle for promoting the cult of Tito or Titoism (Perica 2010; Kuljić 2011), as a means to construct social memory and wield political influence through meticulously choreographed mass spectacles. Evident in Tito's speeches was his special attention to the youth population, whom he regarded as the torchbearers of the revolutionary journey toward societal transformation, providing them with an occasion like the *slet* to celebrate their robust and physically capable bodies. During these performances, he observed his idealized nation from atop the stadium, assuming the role of the foremost ideologue and paternal figure of the nation, esteemed through the grand messages of love and admiration projected by the participants within the stadium. Much like the staged adaptations of chain dances, it remained imperative that the *sletovi* were executed by amateur participants with unknown names, yet engaged in mass participation, free from the necessity of demonstrating flawless choreographic mastery and virtuosity. In addition, the participants were not professionals but rather amateurs from all of the Yugoslav republics, who travelled to Belgrade and participated on a voluntary basis (students were excused from attending classes), guided by the idea that they served as representatives of their respective republics and devoted long hours to rehearsing their regimens prior to the event. The organizers of the *sletovi* changed every year, but they comprised the most notable Yugoslav choreographers, directors, and music composers, who were paid large sums of money to conceptualize and rehearse the performance. The months-long organization of the *sletovi* comprised an industry in its own right, given that it included not only the participants but also managers of hotels, restaurants, and transport companies in Belgrade that accommodated the participants at the expense of the government (Jakovljević 2014). However, as Jasna Žmak points out, detailed information regarding the organization and the financing of the events remains unavailable to the public, along with the lists of songs, folk dance ensembles, sports clubs and bands, and even the names of the choreographers and directors who directed them (Žmak 2020).

Upon a comprehensive examination of a diverse array of *slet* performances spanning the period from the 1960s to the 1980s, it becomes apparent that the ritualistic framework remained consistent, maintaining established protocols and a predefined program agenda, while variations were discernible in choreographic choices. Nevertheless, a persistent feature was the presence of commentators who undertook the task of contextualizing the televised performance for the Yugoslav audience, intertwining political and ideological narratives to reinforce a collective sense of unity and underscore the national significance of the event. In terms of visual symbolism and color utilization, considerable prominence was accorded to the portrait of President Tito, symbolized as the paternal figure of the nation and the recipient of an entire spectrum of ideological messages and choreographed performances, symbolically conceived as a gift for his unwavering dedication to the Yugoslav people. Participants, attired in the tricolors of white, blue, and red, mirroring the Yugoslav flag, orchestrated their collective bodies with precision to project images and symbols that frequently encompassed the star emblematic of the Yugoslav flag, alongside slogans that conveyed messages venerating the president and the nation, exemplified by phrases in the style of "Long live Yugoslavia" or "Long live President Tito" (see Figure 3). It is important to add, however, that these performances were referred to as "exercises" (*vežbe*) rather than "dances". The overall performance was accompanied by propaganda-infused music, either rendered by symphonic orchestras or the Yugoslav army orchestra, or custom-composed popular music executed by renowned artists, occasionally tailored specifically for the event.

The essence of the *slet* primarily entails the orderly progression of participants into expansive formations, characterized by a continuous flux of positions that utilize the human body to craft symbolic messages. When stationary, the participants engage in cyclical crouching and standing, contributing to the visual manipulation of the movement's dynamics. These performers often bear various props, such as weapons, illuminated torches, and flags, or employ hand and leg gestures to execute simple choreographic maneuvers. When observed from an aerial perspective, this orchestrated choreography

achieves synchronization and spectacle that can solely be attained through the collective participation of the masses. As previously mentioned, an integral component of the *slet* is the rendition of *kolo/oro* and other arranged and staged folk dances, executed by dancers adorned in traditional attire representative of the Yugoslav republics. While the musical composers, vocalists, and choreographers vary from one year to the next, they typically encompass esteemed figures within their respective domains, well-recognized by the event's spectators. Constant contributors encompass the Yugoslav army personnel, athletes, sports professionals, gymnasts, dancers, children, youth, and ordinary citizens who personify the nation's collective body. A particularly noteworthy segment of the event involves the formal presentation of the baton to President Tito, accompanied by a speech and a letter, in which the individual delivering the baton, as a representative of the nation, conveys well wishes to Tito and the Yugoslav populace.

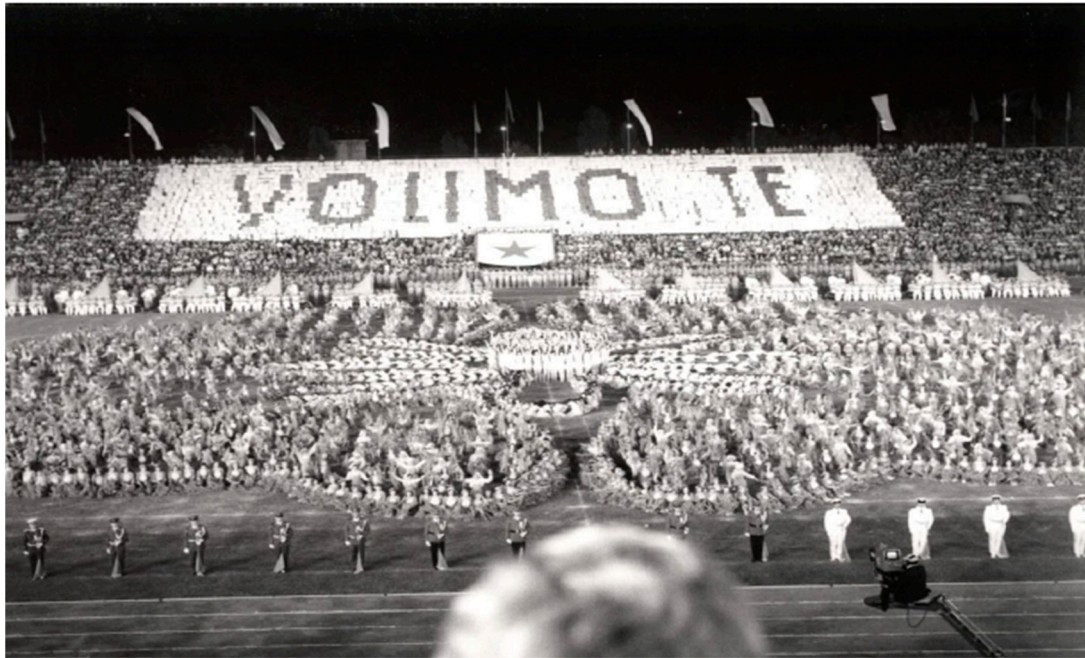

**Figure 3.** President Tito observing the *slet* performance and the message "We love you" in 1972 (source: Museum of Yugoslavia, Photo 1972_490-217).

The stage setting of the *slet* performances at the stadium in former Yugoslavia held profound significance in conveying the intended message and ideology of these grand-scale events. The open layout of the stadium provided the canvas for the involvement of thousands of participants, embodying the vast expanse and inclusivity of the Yugoslav state. Central to the stage arrangement was a focal point where key figures, including President Tito, were typically situated, emphasizing the political and leadership dimensions of the *slet*. The adornment of the stage with an array of flags, banners, and socialist symbols functioned as a visual representation of the state's political ideology, while the presence of flags from all Yugoslav republics underlined the federation's multi-ethnic character. Lighting and audiovisual effects further heightened the spectacle's impact, adding an immersive dimension to the performances, while the audience, comprising dignitaries, political leaders, and citizens, facilitated a collective experience, fostering shared enthusiasm for the event.

Ana Vujanović theorizes that the *sletovi* serves as an illustrative example of social choreography due to their mass participation and repetitiveness, mostly because of their ability to showcase how the ideology is inscribed in the body (Vujanović 2013, p. 22). She contends, however, that the *slet*, as a form of social choreography, does not depict how Yugoslav society appeared but rather how the Yugoslav state presented itself in the public

sphere as a model of a social entity, given that it represents a dynamic system of gestures, which serve as a site of collapse where illegible movements and non-gestures emerge. These elements represent the embodiment of the "choreographic unconscious" of the *slet*, indicating the socialist state's inability to fully align with its own society (Vujanović 2013, p. 22). A similar exploration of the ideology invested in such choreographic patterns has also been explored by filmmaker Marta Popivoda in her documentary feature "Yugoslavia: How Ideology Moved Our Body" (Popivoda 2013). Popivoda underscores the performative nature of politics and the construction of political narratives in socialist Yugoslavia, positioning the *slet* as a carefully orchestrated project aimed to portray an idealized image of the state in which the bodies of the participants are not merely dancing but performing a political script.

In terms of gender politics, the Yugoslav government aimed to promote an ideology centered around the Socialist physically capable body that abandons the hierarchy and the binary of male and female gender in the sense of femininity and masculinity. However, as Slavcho Dimitrov argues, the collective conception of the socialist political body was heavily influenced by the historically shaped characteristics and privileges associated with masculinity, while such preferences in a hierarchical manner are closely linked to the structured categorization and arrangement of bodies based on sexual differentiation (Dimitrov 2014, p. 6). In order to realize the materiality of this philosophy, Yugoslav citizens had to be healthy and fit but most importantly, organized under the idea of the collective in order to better serve the Yugoslav society and influence solidarity, cooperation, and unity, as the opposite of the capitalist emphasis on individualism. Placing emphasis on the importance of physical culture as a distinct aesthetic, this ideology confronted the idea of the Western understanding of competition that was promoted in Western cultures but rather encouraged the collective and voluntary participation of disciplined bodies working in service of the state.

It is evident that these theatricalized performances largely served as a strategy for creating public perceptions of socialist values, beliefs, and political messages, centered around the idea of diversity, unity, self-management, and power to the working people. What is intrinsic to these examples and common to the mass participation in chain dances, is the notion of society as an organism, where members are active participants in the creation and dissemination of socialist values, and the creators of the communal ideological aesthetics. Furthermore, the Yugoslav society and the collective body of the nation is not only depicted through the thousands of participants at the stadium and the participants in the baton race but also through the millions watching through their TV monitors at home. Together, they participate in accomplishing a task of national importance, that is, glorifying the socialist society and pledging allegiance to their leader by observing their fellow community members performing the idealized community. These events, in return, were supposed to provide both participants and observers with a sheer sense of happiness, gratitude, and communal love[2].

The performance of the *slet* resembles the performative realm of ideology that Hewitt explores in his work, one that induces discipline and order manifested in the participation of bodies that produce an economy of experience as an affective ordeal. Furthermore, as Dimitrov argues, the collective "We" in social choreography manifests as an extensive network of transmissions and physical interactions, characterized by motor and emphatic contagions, fostering a sense of intoxication and ecstasy that flows among and between bodies (Dimitrov 2014, p. 52). By participating in the choreography, participants offer their bodies to the state but this offer is then rewarded with the opportunity to receive back, in this case, with the feeling of sharing, the sensation of experience materialized only through mass participation of bodies acting as one. As Erin Manning contends, the body does not move into space and time, but it creates space and time that would be impossible before movement (Manning 2007, xiii). Hence, to think and act politically, in this sense, is to become aware of how bodily movement creates a sense of stability,

which the state apparatus perceives is of utmost importance in achieving the goals of unity and togetherness.

## 4. Choreographies of Protest

The last *slet* was performed in 1988, although in a different manner than previous occasions. Namely, a central position was given to a contemporary dancer Sonja Vukičević, who performed a piece entitled "1988". This peculiar decision to feature a single body as the main focus of the performance was historical, first of its kind, and unfortunately, the last. The performance mediated images of an individual agonized and torn by individual problems, expressed through relentlessly running in concentric circles, surrounded by the static bodies of other members of the community. In a way, this performance symbolized Yugoslavia's downfall and disintegration into independent states and transition to capitalist societies. Furthermore, Vujanović points out the observations made by commentators in regards to the *kolo* performances. They noticed that instead of dancing together, as they used to, dancers performed separately to the tunes of their national melodies. As the commentators concluded, "it looks like we are not united as we should be" (Vujanović 2013, p. 25).

The disintegration of Yugoslavia, a topic extensively analyzed by various scholars (Benson 2001; Wilmer 2002; Kecmanović 2002; Hudson 2003), was primarily precipitated by the weakening of the socialist framework amid a surge in nationalism and separatism. This turmoil eventually erupted into violent conflicts and episodes of ethnic cleansing. The declaration of independence by Slovenia, Croatia, and Macedonia in 1991 marked the initial fractures, followed by Bosnia and Herzegovina in 1992, and later, Montenegro and Serbia in 2006, with Kosovo following suit in 2008. Subsequent to the dissolution of Yugoslavia, the post-Yugoslav nations embarked on nation-building endeavors and the construction of distinct national identities and a transition towards capitalism. However, certain Yugoslav ideals, especially in relation to the performance of the collective body, did not disintegrate but were reintroduced, as evident in anti-government protests in Macedonia and Serbia.

In recent years, many of the people within the former Yugoslav republics, as shown from the examples of Macedonia and Serbia, have publicly expressed their lack of support towards their governments in the form of mass protests. For instance, Macedonia bore witness to a significant event known as "The Colorful Revolution", unfolding from 12 April to 20 July in 2016, in response to a range of criminal activities, including wiretapping, police brutality, acts of violence, and widespread corruption. Before the revolution, the Macedonian arts and performance scene had been characterized by a preoccupation with themes aimed at reinforcing national identity and producing propaganda materials that extolled the Macedonian people and their history. As Tihomir Topuzovski notes, the dominant conservative and nationalistic party in Macedonia perceived culture as a potential instrument for shaping collective memories, patriotic motifs, slogans, symbols, and monuments. This approach elevated culture to the point of idolatry, as part of an endeavor to reconstruct the historical narrative and give it meaning in the contemporary context. In line with this, nationalistic art in Macedonia typically exalted declared facets of national and romanticist spirit, characterized by elaborate ceremonies and a strong emphasis on re-establishing national identity within the realm of post-socialist material culture (Topuzovski 2017, p. 15).

In opposition to these ideas, the revolution garnered support from the European-Union-oriented, social-democratic party, SDSM. Over an eleven-year period, during the reign of the previous right-wing political party VMRO- DPMNE, artists found it increasingly difficult to secure opportunities unless their work adhered to a nationalistic character or openly endorsed the ruling political party. In response to these limitations, protesters took a novel approach, using vibrant paint to adorn the newly constructed faux-baroque and neo-classical buildings of the capital city of Skopje with a rich array of colors, ultimately forming a visual representation of the rainbow. Amidst the backdrop of the Colorful Revolution, wherein the protestors unequivocally voiced their opposition to political and artistic censorship while calling for pro-European political and societal changes, there existed

another faction that organized counter-protests, often referred to as anti-protests. These demonstrations aimed to express unwavering support for the ruling party, which reciprocated with offers of employment and assured avenues for career advancement—a privilege primarily reserved for political loyalists. Irrespective of their willingness to partake, many were transported to the capital city, where they were instructed to abandon their jobs and assemble in front of the parliament building, symbolically clapping in solidarity with the ruling party. The anti-protests were orchestrated by a coalition known as "For Macedonia Together" (*Za Makedonija Zaedno*), generously backed by the government, who opposed the inclusion of ethnic Albanians in the Parliament and the European Union aspirations.

In neighboring Serbia, protests and demonstrations unfolded primarily directed against the government, led by the Serbian Progressive Party (SNS), a right-wing political entity, and President Aleksandar Vučić. The protests can be traced back to the early 2010s when the SNS, led by Aleksandar Vučić, began gaining political influence. Vučić, who had previously been a member of the Serbian Radical Party with strong nationalist leanings, adopted a more moderate political image while presenting himself as a pro-European integration figure. However, as SNS consolidated its rule, concerns arose regarding the erosion of democratic institutions, media freedom, and the concentration of power in the executive branch. The first notable wave of protests erupted in April 2016 after the presidential election in which Aleksandar Vučić was elected president. Concerns regarding election irregularities, media bias, and the dominance of SNS in the political landscape fueled the discontent. The protests were largely organized by civil society organizations and opposition parties, and they were predominantly peaceful.

Similar to the Macedonian protests in response to the faux-Baroque reconstruction of the capital, in 2014, Serbs protested against the Belgrade Waterfront Project aimed at transforming the Sava River waterfront into a modern, high-end urban area with luxury residential and commercial properties, including the construction of Belgrade Tower, a massive skyscraper that was promoted as a symbol of modernity and urban development. The demonstrators opposed what they perceived as a lack of transparency in the project's planning and decision-making process. They argued that the project disproportionately favored private developers at the expense of public interest and that it would exacerbate issues such as housing affordability and environmental sustainability. Demonstrators saw the project as emblematic of the government's alleged authoritarian tendencies, corruption, and disregard for public opinion. The project became a symbol of what they considered to be an undemocratic and opaque governance style.

Protests escalated in December 2018, with the most significant demonstration taking place on 8 December. These protests, known as the "1 in 5 Million" movement, were sparked by an attack on the leader of the opposition party "Levica" (the Left), Borko Stefanović. Demonstrators demanded media freedom, an end to political violence, and the establishment of independent institutions. Most recently, in 2020, the Serbian government faced additional challenges when it imposed strict anti-COVID measures. Protests erupted in July 2020, triggered by concerns over the government's handling of the pandemic, including allegations of mismanagement and lack of transparency. The initial manifestation of dissent originated from students residing in dormitories, who, in an act of defiance, convened in proximity to the Assembly of Serbia. The catalyst for this mobilization was the slated closure of dormitory facilities as a preventive measure against the propagation of COVID-19. Subsequent to swift intervention by governmental authorities, the directive to shutter the dormitories was rescinded. Nevertheless, the demonstrations persisted, witnessing expanded participation encompassing a broader demographic of civic constituents. Citizens gathered to express their dissatisfaction with the government's actions and accused it of exploiting the pandemic to further tighten its grip on power, in, what could be explained as a radical protest in which the police used violence and tear gas, and arrested many of the protesters.

What was common in the Macedonian and Serbian protests were concerns about democratic backsliding, media censorship and control, allegations of electoral fraud, and

growing authoritarian tendencies within the government. Additionally, economic issues, including high levels of unemployment and economic inequality, contributed to the public's discontent. The governments responded to the protests with a combination of repression, propaganda, and political maneuvering. For instance, in Serbia, following the incorporation of *kolo* dancing during the protests, there followed a visible decrease in the fervor of the demonstrations. The assembly of participants decreased progressively, ultimately culminating in the fall of protest activities. Furthermore, the media opportunistically seized upon this development, framing the protests within the public discourse as a triumphant gathering. This narrative was accentuated by media headlines that portrayed these occasions as joyous acts of celebration. In addition, the government-controlled media sought to delegitimize the protests by portraying them as the work of foreign agents.

However, these protests did not go without the participation of ideology-free bodies and the imbued remnants of the socialist past. What was different, however, was that these masses of people were not obliged by the state to embody their support for their leaders and the nation as a whole. Rather, the people in attendance participated voluntarily, driven by the desire to return to a democratic society—one they had imagined and awaited for a long time. Chain dances, such as the *kolo* and *oro* that I previously mentioned, accompanied by traditional and patriotic music, took prominent place in the protests, at times spontaneously, while in other instances, on demand by the ruling political party. However, these dances were performed mainly by anti-protesters, that is, people who took to the streets to demonstrate their support for the governments and oppose the current protesters. For instance, in the Macedonian anti-protests, spontaneous *oro* performances unfolded organically, but these were complemented by meticulously orchestrated presentations held right in front of the parliament building. Amateur dance ensembles and The National Ensemble "Tanec" took the stage in these symbolic performances, elevating the significance of certain dances, notably, *Kopachkata*[3], a social dance inscribed on the UNESCO List of Intangible Cultural Heritage of Humanity, and *Teshkoto* (See Willson 2014), a male dance often hailed as the emblematic prototype of Macedonian dance forms. Both dances, due to their historical and cultural significance, have become potent symbols of Macedonian identity, achieving both national and international acclaim, so it was only natural that they were performed on occasions that called for the invocation of national symbols.

Aware of the power of using national images, symbols, and performances of national character, as frequently conducted by the right-wing government during the apex of the Colorful Revolution, the Social Democratic party (now in power) made multiple attempts to politicize the revolution. In doing so, they employed the poet Blazhe Konevski's poem about *Teshkoto* in a promotional video as part of a political campaign. The video aimed to poetically and visually illustrate the oppression allegedly perpetuated by the right-wing political party. This marked a shift from the dance's historical use, where it previously symbolized oppression and suffering under Ottoman colonization. In this instance, it was repurposed in a political campaign video to narrate the perceived hegemony imposed by the ruling political party. Eleven years prior to the revolution, during the rule of the VMRO dictatorship, the dance was once again exalted, playing a key role in a nationalistic agenda. An emblematic instance was the unveiling of a new statue of *Teshkoto* in the city center, part of an ambitious project aimed at transforming the capital into a European metropolis. Throughout the daily marches of the Colorful Revolution, prominent figures in Macedonia's arts and entertainment sphere, including directors, actors, and dancers, used Western and foreign music, as they deemed performances of traditional music and dance as nationalistic, backward, and anti-European. In a poignant evolution, chain dances, once utilized in accordance with Yugoslav socialist ideals of brotherhood and unity, were now harnessed as potent instruments for expressing anti-nationalistic sentiment.

In Serbia, instances of dancing *kolo* have been evident in mass demonstrations even prior to the events that I refer to. The student protests in Yugoslavia in 1968 were part of the global wave of movements advocating for social and political change, commonly associated with the larger context of the May 1968 protests in Western Europe. In Yugoslavia,

students voiced dissatisfaction with the political system, expressing a desire for greater political freedoms, democratic reforms, and increased individual liberties. During the demonstrations, students organized collective kolo dances in public spaces as a form of nonviolent resistance. The circular dance, with participants holding hands and moving in unison, became a powerful visual representation of communal strength and shared purpose. By integrating *kolo* dance into the protests, the students aimed to convey a message of cultural pride and national unity while advocating for political change. In addition, as Lynn Maners notes, *kolo* was performed during NATO's bombing of Belgrade in 1999, by local Belgrade women dressed in traditional costumes (Maners 2006, p. 76). The tactic employed by the Serbian people was to use their bodies as a shield, possibly led by the ideological matrix that was still interwoven in their corporeal memory, to defend the city infrastructure that were the targets of NATO's bombings. In connection with the recent protests I have mentioned, supporters of Vučić's government, who would also gather in large numbers in order to demonstrate their support for the president, would spontaneously break into *kolo* dances during demonstrations, using these traditional forms of expression to convey messages of support for the government. By incorporating *kolo* dances into the protests, participants reinforced their cultural identity and resilience, while the choice to connect to their national roots and folklore conveyed a message of preserving Serbian culture in the face of what protesters perceived as threats to their cultural heritage. The aim here was not to perform *kolo* as the emblem of the culture of the working people, nor as a heritage that was praised because of its ability to educate society on the country's rich culture and history. Rather, it was used to reassure its citizens of their distinct Serbian nationality in occasions where participants in mass events felt that it might be threatened.

What these examples indicate is the power of the corporeal dimension of ideology that becomes intrinsic to the concept of body politics. As argued, remnants of the socialist past have become evident in the utilization of social dance and choreography in situations that require mass participation. As Ana Hoffman contends, many of the scholars who argued that socialist culture was a forced form of cultural creation found themselves taken aback by the observation that the transition to post-socialism did not bring about a dramatic shift in the cultural dynamics of former socialist nations. Contrary to their expectations, the collapse of "socialist regimes" did not lead these societies to promptly erase their "totalitarian past" but, in fact, there remained a significant degree of continuity with earlier periods, especially in the realm of culture (Hofman 2011, pp. 238–39). Essentially, the continuum of certain state-made aesthetics and tactics aimed to choreograph, discipline, and perform society were now used within a different context. This process has been the subject of theorization by Joseph Roach, who conceptualizes it as surrogation, that is, the intricate interplay of memory, performance, and substitution within cultural contexts where cultures perpetuate and reinvent themselves as they grapple with disruptions in social relations. In response to these disruptions, individuals within a community engage in a process of seeking suitable replacements or substitutes to fill the resulting voids, within the framework of selective, imaginative, and often unpredictable collective memory (Roach 1996, p. 2). Through a form of collective memory, the society engaged in protest turns to dancing *kolo/oro* as a surrogate for expressing cultural identity while engaging in a dynamic act of resistance.

Spontaneous participation in social dances embodies the enduring legacy of a shared cultural and communal experience of the socialist past, as it embodies the historical bond that unites people who once identified as part of a broader Yugoslav society. In such instances, it is evident that the aesthetic is inseparable from the political, as physical and political dimensions are intertwined through mass bodily participation that impacts societal perception. In the realm of both performance and protest, the intrinsic ephemerality poses challenges and underscores the sense of lack or incompleteness. Randy Martin's perspective (Martin 2011) on dance and protest as phenomena that share an uncertain and ephemeral quality highlights their transient nature, which often results in a feeling of unfulfillment. The utilization of *kolo/oro* in modern protests seems to transcend this ephemerality, offering

a connection to the past and instilling a sense of cultural continuity despite the protest's potentially fleeting impact. This persistence of a cultural symbol serves as a powerful medium for political expression and mobilization, with *kolo* and *oro* acting as significant bridges between the past and the present.

The mentioned examples of mass protests can be effectively theorized as a form of social choreography, encapsulating the intricate interplay of collective movements, gestures, and symbolic actions within a sociopolitical context, and offer a deeper understanding of how mass protests embody a structured, performative, and symbolic dimension in the realm of social and political activism. Social choreography, as Hewitt has outlined, extends the concept of choreography beyond the traditional confines of dance and performance to encompass the organization and coordination of bodies in public spaces, often serving as a means of political expression. In the context of mass protests, it underscores how demonstrators collectively move, assemble, and engage in various actions to convey their shared grievances, demands, and visions. Furthermore, social choreography emphasizes the choreopolitical nature of mass protests, highlighting how these events serve as platforms for the negotiation and contestation of power and authority, as protesters engage in embodied acts of dissent, contesting established social and political norms. Their movements, gestures, and symbolic actions are imbued with political and ideological meaning, challenging existing power structures and ideologies. Affect, expressed through dancing and marching, plays a central role in shaping the emotional and sensory experiences of demonstrators, evoking notions of solidarity, empowerment, or outrage, which are integral to the choreographic nature of these events.

## 5. Conclusions

In the course of this study, I have conducted an examination of the confluence of choreography and political ideologies, particularly in the context of mass choreographic spectacles and movements within the former Yugoslavia and the subsequent post-Yugoslav national states. The core premise revolved around the creation and representation of a collective community, intricately intertwined with political ideologies, which harnessed folklore, physicality, and protest as means of either reinforcing or dismantling state formations. Within this ideological landscape, mass participation took on the role of symbolizing the virtual socialist body and manifesting the ideals of brotherhood and unity, which were central to the Yugoslav ideology. Notably, this decentering of individualism posed challenges to the promotion of private identities, as they risked being perceived as a threat to the prevailing ideological framework. Instead, the emphasis was placed on cultivating a balanced array of identities, encompassing ethnicity, culture, religion, and physically capable bodies, united under the overarching goal of symbolizing and publicly representing Yugoslav socialist society.

Taking a cue from Andrew Hewitt's concept of social choreography as a framework for analyzing choreographic mechanisms in non-dance societal practices, this study offers valuable insights into the choreographing of society in alignment with distinct ideologies, notably those centered on socialism, unity, togetherness, and the commune. The examination of performance as a medium for expressing ideologies illuminates how collective and mass dance performances served as vessels for embodying Yugoslavia's focus on the commune over individualism. The state's prioritization of these dances as national assets aligned with broader Yugoslav national policies, emphasizing the concepts of brotherhood and unity through shared bodily movements and expressions. Amid these discussions, the significance of delving into these topics becomes increasingly evident.

Analyzing how ideologies are embodied through choreography provides a unique lens through which to understand the performative nature of protests and mass gatherings. This exploration reveals how societal dynamics are intricately choreographed, and the engagement of the collective body in public displays of solidarity is essential to grasp the nuances of social movements. By examining how choreography is employed to uphold, reinforce, or challenge the ideological foundations of states, we gain a deeper comprehension

of how societies have been historically choreographed to conform to particular worldviews and, conversely, how societies can also wield choreography as a tool of dissent. Importantly, the discourse surrounding choreography's role in shaping societies and protests is not confined to the realms of dance or performance studies but resonates across various academic disciplines. It underscores the complex interplay between physicality, collective identity, and the articulation of political ideologies in societies. This interdisciplinarity becomes increasingly critical in contemporary academia, where the lines between various fields of study are often blurred, reflecting the interconnectedness of the human experience. The choreography of societies and protests invites us to explore the multifaceted aspects of human expression, agency, and resilience, making it an essential and timely subject of academic inquiry, particularly in an era marked by political and social upheaval.

**Funding:** This research received no external funding.

**Data Availability Statement:** No new data were created or analyzed in this study. Data sharing is not applicable to this article.

**Conflicts of Interest:** The authors declare no conflict of interest.

## Notes

[1] The opposite aesthetic of *stilizacija* is a choreographic process that places emphasis on "authenticity" or *izvoren folklore* (authentic folklore) through which the chain dances, when arranged for the stage, only undergo small modifications in relation to shortening the duration of the dance, accelerating the tempo, and including some small acting parts and/or narratives in order to contextualize the dance as folklore. Other than these changes, the dance and the dancing are supposed to be resemblant to performances in social setting, while choreographers were advised not to change the dance steps and movements so the dance will appear "original" and "authentic".

[2] For similar discussions, see (Foster 2019).

[3] https://ich.unesco.org/en/RL/kopachkata-a-social-dance-from-the-village-of-dramche-pijanec-00995 (accessed on 15 December 2023).

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
