# Peer review of "How to Choreograph a Socialist Society?"

_arts, 2023_

Round 1

Reviewer 1 Report

Comments and Suggestions for Authors

The article “How to choreograph a socialist society?” presents a compelling introduction to how dance and choreography were strategically used in mass performances during socialist Yugoslavia and in post-Yugoslav protests. The strongest parts of the article are descriptions of choreography and the discussion of how the post-Yugoslav uses of kolo/circle dances related and diverged from the socialist Yugoslav context. Overall, however, there is an excess of historical summary and summary of dance/ideology theory. Lengthy summaries of regional history both pre- and post-socialist Yugoslavia distract from the central argument about dance and choreography. For example, lines 577-680 make no mention of dance and it is not clear why the history is being presented. Most of the mentioned nuances of the history are irrelevant to the later discussion of the dances in the protests. Additionally, and perhaps most importantly, the dances feel abstract, rather than concrete. The article seems to present dance as an idea, rather than something that is performed by real people in real spaces. This tendency toward abstraction detracts from the author’s central argument and obscures the significance of dance as an embodiment of ideology.

There are a few cases where the author alludes to an idea, but does not expand on it. For example, the author mentions that the article is part of a larger recent trend in studying the choreographing of Yugoslav society (lines 24-25), but citations are needed to support this. Situating the present article in context of these other studies would benefit the argument and hopefully make clearer what the specific contribution of this article is. The author also regularly references that mass choreography was an embodiment of “socialist values,” but what is meant here by socialist values? The redistribution of wealth? The repossession of the means of production by workers? More specificity on the socialist aspect is needed. Socialism is not a mere synonym for totalitarianism or collectivism, for example.

The largest issue with the article is the lack of discussion of bratstvo i jedinstvo. The author mentions this as the primary ideology that was expressed through mass choreography in socialist Yugoslavia, however, the author does not provide a thorough explanation of this complex concept and its significance in dance (and more broadly). Historical context for the idea as well as a critical discussion would benefit the article tremendously. The gendered aspect in particular needs to be expanded upon, especially as the author mentions the significance of gender in kolo (lines 251-255). Slavcho Dimitrov’s work on gender is referenced later, but there should be a direct discussion of the masculinist dimension of bratstvo i jedinstvo (some Yugoslav and post-Yugoslav feminist scholars could be cited here as well). This does not need to be a Western feminist condemnation of the male-ness of “bratstvo,” but rather some discussion of brotherhood is needed. What is meant by “brotherhood” in socialist Yugoslavia? Brotherhood is an element of many socialist ideologies, so the author could connect Yugoslav bratstvo i jedinstvo to other socialist models of brotherhood and friendship.

In addition to the article benefitting from theoretical specificity, the article would also benefit from more specificity when discussing sletovi. Choosing a case study event from a single year would help make clear how these theoretical and ideological dimensions were actually expressed on the ground, rather than solely in abstract choreography. As a reader I was left wondering who the choreographer(s) of the sletovi were, how they were choreographed/rehearsed, whether the performers professionals or amateurs, whether the performers were paid, and how the performances were received/remembered. How was the ideological content of the dances confirmed or contested by party leadership? Are there examples where the choreography was rejected or modified in response to criticism? I felt that I only gained an abstract understanding of sletovi and the role of dance in them (dance was not the only component, how did it fit with the other elements?). I wanted more concrete examples. If these dances were embodiments of ideology, how did that ideology impact the bodies and lives of the actual performers? Are there testimonies or interviews that could be used as supporting evidence?

Overall, with some improvements this article would be a great contribution to our understanding of state performances in socialist Yugoslavia and the continuum between superstructural ideology and people’s living bodies.

Comments on the Quality of English Language

Minor adjustment of articles, commas. 

Author Response

Thank you very much for the very useful suggestions and comments that have helped me produce a revised version of my paper. In response to your comments:
1. I have paired down and shortened the theoretical discussions about choreography and ideology. However, having in mind that many readers might not be familiar with the existence of Yugoslavia, or its history, I decided to keep certain historical information to contextualize the development of the Yugoslav socialist ideologies.
2. Regarding the comments that the dances feel abstract, please keep in mind that I am analyzing a dance production that spans around fifty years, and hence I am not focusing on specific performances. I am rather explaining how this dance production was motivated by state ideology, mostly through a historical perspective. I present the ideas and motivations regarding the dance and choreographic productions, while historical information might help the reader to contextualize the developments throughout history, in comparison with recent protests and the break up of Yugoslavia.
3. I have added references to which authors have been studying the topic of choreographing society, as suggested, and I have exemplified what I mean by "socialist values" upon the mention of the term.
4. I have substantially expanded my discussion of brotherhood and unity, including its historical context, comparison to other such ideologies, and the gendered aspect, including the works of other authors who have written on this topic.
5. I have included answers to your questions about sletovi and I have mentioned that much of the information about them remains unavailable to the public (including funding, rehearsals, names of participants and choreographers, etc). However, I decided not to focus on a specific slet, but I am rather analyzing what was common in all performances of sletovi. As an example, I point out the last slet, which was different from the others, and hence, deserves a specific mention that also initiates the discussion of the break up of Yugoslavia.
Please see the attached revised version of my article, and once again, thank you very much for the useful comments and suggestions. 

Reviewer 2 Report

Comments and Suggestions for Authors

The paper represents an interesting approach to analysing important socio-political topics in which dance and choreography play a prominent role. The author uses relevant theoretical frameworks, literature and materials to reach meaningful conclusions. The author can find several comments in the attached PDF that can help him in the following reflections or the eventual addition of some parts of this paper.

Author Response

Thank you very much for the useful comments and suggestions. I have incorporated all of the suggestions that you mentioned and they will be visible in the next draft of the text which will also contain changes suggested by the other reviewers. Thank you once again!

Reviewer 3 Report

Comments and Suggestions for Authors

There is a typo in line 52. Please make a general revision of the text to avoid further possible typos.

I would number/list the research questions on page 2 (lines 67-72) with a number or something similar so that they can be better focussed when reading.

If you cite an author with the page reference, then I expect a citation in the preceding text (see for example: lines 95, 98 and following).

Another issue related to citations is that you cite the edition you are reading, and that is correct, but in some cases you put the year of the first edition in the parenthesis and in others you do not. I think it is good practise to cite the year of the edition you are reading AND the first edition in square brackets (for example, in the case of M. Foucault's texts). However, please use only one unambiguous way.

Please align the photos and illustrations with the rest of the text, as some have already done.

A blank space on p. 21 should be deleted.

Author Response

Thank you very much for the useful comments and suggestions! I fixed all of the typos and cleared the unnecessary empty spaces. I added the original years of publications of the works I cite (Marx and Foucault) and I numbered my research questions. I have also aligned the photographs with the text. As for the citations with page numbers, to my understanding, authors can paraphrase the text and cite the year and the page number, without having to directly copy the original text and provide it as a direct citation. Please see the document attached. Thank you once again!
